# INTEGRATING PLANNING INTO SINGLE-TURN LONG-FORM TEXT GENERATION

## ABSTRACT

Generating high-quality, in-depth textual documents, such as academic papers, news articles, Wikipedia entries, and books, remains a significant challenge for Large Language Models (LLMs). In this paper, we propose to use planning to generate long form content. To achieve our goal, we generate intermediate steps via an auxiliary task that teaches the LLM to plan, reason and structure before generating the final text. Our main novelty lies in a single auxiliary task that does not require multiple rounds of prompting or planning. To overcome the scarcity of training data for these intermediate steps, we leverage LLMs to generate synthetic intermediate writing data such as outlines, key information and summaries from existing full articles. Our experiments demonstrate on two datasets from different domains, namely the scientific news dataset *SciNews* and Wikipedia datasets in *KILT-Wiki* and *FreshWiki*, that LLMs fine-tuned with the auxiliary task generate higher quality documents. We observed +2.5% improvement in ROUGE-Lsum, and a strong 3.60 overall win/loss ratio via human SxS evaluation, with clear wins in organization, relevance, and verifiability.

## 1 INTRODUCTION

Large language models (LLMs) have achieved remarkable progress in various text generation tasks, ranging from creative writing to summarization and dialogue generation (Li et al., 2024; Gao & Callan, 2021). However, generating high-quality, coherent, and substantive long-form documents, such as academic papers, news articles, and books, remains a significant challenge (Sun et al., 2022; Tan et al., 2020). Existing work mostly tackles this challenge by sequentially prompting LLMs to write a small segment of the document at each call and integrating the outputs into the final long-form document (Tan et al., 2020; Yang et al., 2022; Shao et al., 2024). Such systems usually require additional components to ensure the coherence or consistency of the entire document (Cho et al., 2014; Xu et al., 2019).

In this paper, we present a novel approach that directly fine-tunes LLMs to generate long-form documents in one single call. This approach enables us to fully leverage the token-level attention mechanism in the decoding process, thereby ensuring the coherence and consistency of the generated document. It also significantly reduces the complexity of the system during serving and deployment to write long-form documents.

Writing high-quality long-form documents often involves a pre-writing stage (Rohman, 1965) where authors outline the structure, develop key arguments and plan for the overall flow of the document. This additional stage enables the writers to simplify the tasks into manageable sub-tasks, similar to the idea of Chain-of-Thought (CoT) in multi-step reasoning (Wei et al., 2022). In fact, this stage is often included in the design of multi-stage long-form document writing systems (Yang et al., 2022; 2023; Shao et al., 2024) through multiple rounds of prompting.

Motivated by this idea, we propose to integrate this pre-writing stage into our development of the single-turn LLM writer. Specifically, we introduce a series of auxiliary training tasks to endow LLMs with the skills to plan and structure long-form documents before generating the final full article. For example, one auxiliary task could involve providing the LLM with the writing context as input and expecting it to produce an outline with key insights as the output. Another auxiliary task can present the LLM with the writing context and an outline, with the goal of generating the complete article. We argue that fine-tuning the LLM writer with a mixture of writing tasks, coupled with guidance at

varying levels of granularity, can enhance the model's ability to produce long-form documents that are inherently well-structured and coherent.

While it is relatively easy to obtain sufficiently large corpora of full articles for supervised fine-tuning, obtaining intermediate writings such as article outline and key points directly from human writers is considerably more challenging as these are typically not well documented and made public. To address this, we leverage the few-shot capabilities of LLMs to generate synthetic intermediate writings from full articles, along with the original document structure when available. Note that, generating a concise summary, excerpt, or outline from a full-length, detailed article is much easier than doing the reverse. Therefore, it becomes rather manageable to create abundant intermediate-writing data for the purpose of fine-tuning LLMs towards learning to plan for writing full articles.

Our experiments on multiple datasets demonstrate that LLM writers trained with the auxiliary tasks generate higher quality long-form documents in a single pass, even when the final inference task does not prompt the model to produce intermediate planning steps.

Our main contributions are summarized as follows:

- We propose a novel approach that directly fine-tunes LLMs to generate the entire long-form document in a single pass, simplifying the generation process and enhancing coherence.
- Inspired by human writing practices, the proposed framework incorporates the pre-writing stages by introducing auxiliary training tasks that teach the LLMs to plan and structure documents before generating the final text.
- To overcome the challenge of limited training data for intermediate writing steps, we leverage LLMs' ability to generate synthetic summaries, outlines, and key information from existing full articles. This innovative approach unlocks a vast new source of training data for LLM planning.
- Our extensive experimental results demonstrated the effectiveness of the proposed approach on multiple datasets, showing that LLMs fine-tuned with the auxiliary tasks produce higher quality, more coherent long-form documents in a single pass.

## 2 RELATED WORK

**Planning.** Our work contributes to the field of planning in long-form text generation. Humans typically simplify complex tasks into manageable subtasks, a method mirrored in recent approaches employing large language models (LLMs) for planning. Techniques such as Chain of Thought (CoT) (Wei et al., 2022), Zero-shot-CoT (Kojima et al., 2022), Self-consistent CoT (CoT-SC) (Wang et al., 2022) guide LLMs through sequential reasoning by utilizing intermediate reasoning steps. More advanced methods, like Tree of Thoughts (ToT) (Yao et al., 2024), GoT (Besta et al., 2024) enhance these strategies with tree-like and graph-based reasoning structures, respectively. Additional methods, including RePrompting (Xu et al., 2023d) and ReWOO (Xu et al., 2023a) ensure planning accuracy by integrating observational data and using corrective prompting HuggingGPT (Shen et al., 2024) further decomposes tasks into sub-goals solved through iterative LLM interactions, contrasting the one-shot approach of CoT and Zero-shotCoT. Despite these innovations, specialized zero-shot plan generation for specific domains remains to be challenging, addressed by models like LLM+P (Liu et al., 2023) , LLM-DP (Dagan et al., 2023), and CO-LLM (Zhang et al., 2023). Built upon previous works, our approach trains models specifically to enhance planning within the domain of long-form text generation.

**Long-form text generation.** Long-form text generation and question answering (LFQA), which require maintaining coherence over extended texts, remain highly challenging for large language models (LLMs), as evidenced by numerous studies (Fan et al., 2019; Xu et al., 2023c; Krishna et al., 2021; Nakano et al., 2021; Su et al., 2022). Tan et al. (2020) introduced a progressive generation technique utilizing pretrained language models to enhance the creation of extended narratives. Xu et al. (2019) explored discourse-aware neural extractive text summarization, essential for maintaining logical flow and thematic consistency in long documents. There have also been efforts to generate Wikipedia articles, as documented by Banerjee & Mitra (2016); Minguillón et al. (2017); Liu et al. (2018), along with recent advancements by Fan & Gardent (2022). Balepur et al. (2023) developed the Imitate Retrieve-Paraphrase framework to enhance expository writing at the paragraph level,

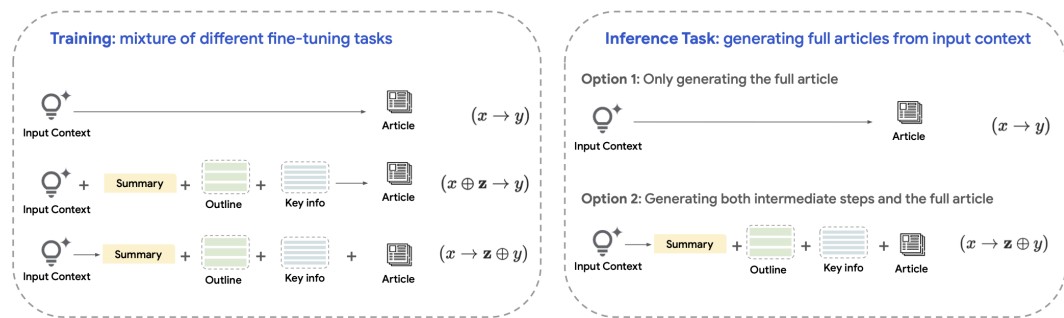

Figure 1: Input/output of training and inference tasks.

particularly focusing on the integration of information from diverse sources. Our research introduces a novel method that leverages organic long documents to create intermediate text generation plans. This approach trains models to enhance their abilities not only in generating text but also in generating and adhering to these structured plans, thereby distinguishing our method from previous work. The use of the Retrieval-Augmented Generation (RAG) technique is outside the scope of our current discussion, although our method is compatible with RAG applications.

## 3 PROBLEM SETUP

**Long-form text generation.** Given an input context $x_i$, which can either be the source academic paper for SciNews generation or a sentence prompt (e.g., "generate wiki page about {topic}" for Wikipedia page generation), the objective is to fine-tune an LLM to generate a long-form article $y_i$ from $x_i$.

The dataset $D$ used for fine-tuning is structured to align with this objective. It consists of pairs of input contexts and their corresponding final documents. Formally, the dataset $D$ is defined as:

$$D = \{(x_1, y_1), (x_2, y_2), \ldots, (x_n, y_n)\}$$

where each tuple $(x_i, y_i)$ represents an input context $x_i$ and the final document $y_i$.

**Intermediate steps.** To reduce the cognitive burden on LLM when generating a full article directly from the input context, intermediate steps are often introduced, gradually leading from the input context to the final full article. In particular, for each input context $x_i$ and its corresponding article $y_i$, the intermediate steps $\mathbf{z}_i$ can either be a set of distinct pieces of information (e.g., summary, outline, key information) or a sequence of information where each element $(j + 1)$-th is dependent on the preceding element $j$. We denote $\mathbf{z}_i$ as:

$$\mathbf{z}_i = \{z_{i1}, z_{i2}, \ldots, z_{ik}\}$$

The overall idea is that the auxiliary information provided at these intermediate steps $\mathbf{z}_i$ can effectively guide the LLM to gradually approach the final target article $y_i$ from the potentially abstract or noisy input context $x_i$. There are multiple ways to instantiate the concrete content of $\mathbf{z}_i$. For example:

1. **Linear:** One can decompose the full article writing task into writing one section at a time. In this approach, the first intermediate step $z_{i1}$ would be the first section; the $j$-th intermediate step $z_{ij}$ would be concatenating one more section to the $(j - 1)$-th draft.

2. **Top-down:** Another common strategy is to use intermediate steps from the most abstract outline to gradually more detailed content. For example, the first intermediate step $z_{i1}$ can be an abstract outline only with the title of each section of the article. Each subsequent intermediate step would gradually elaborate on the content within each section. This kind of intermediate steps are not usually readily available, and may need to be constructed from the full article, which we will describe later in this paper.

Note that many existing works aim to developing multi-turn LLM writers, which involves a series of tasks such as generating $z_{i1}$ from $x_i$, then $z_{ij}$ from $z_{i(j-1)}$. However, this is not the focus of our

paper. In this paper, as shown in Figure 1, we introduced intermediate steps to create different training tasks for fine-tuning LLMs. Our final objective remains to generate the full article $y_i$ from the input context $x_i$ in a single turn. During inference, the input is always only the input context $x_i$. The model can either generate only the article $y_i$ or both the intermediate steps $\mathbf{z}_i$ and the article $y_i$, as long as the article can be easily extracted from the complete model output through post-processing.

## 4 METHODOLOGY

Our proposed framework addresses the challenge of generating long-form text with LLMs by utilizing the inherent structure of documents. Specifically, we construct intermediate steps $\mathbf{z}_i$ based on this structure to guide the generation process. The key insight is that the inherent structure, such as an article's outline, provides crucial guidance for organizing the generated article $y_i$.

Within this framework, we construct the intermediate steps $\mathbf{z}_i$ by extracting the implicit structural information inherent in a formulated document and then constructing a hierarchical writing plan accordingly. These intermediate steps then serve as a foundation for fine-tuning the LLM, focusing on tasks that emphasize structural understanding and plan interpretation. At inference time, the fine-tuned LLM, equipped with enhanced structural and plan interpretation capabilities, generates the final long-form text $y_i$ in a single pass. The specific components of this framework are discussed in the following sections.

### 4.1 WRITING PLAN AS INTERMEDIATE STEPS

Writing high-quality long-form documents typically begins with a pre-writing phase, where authors establish the structural framework, develop key arguments, and formulate the overall trajectory of the narrative. In this work, we define the pre-writing stage as a series of intermediate steps essential for generating long-form documents. Specifically, without loss of generality, we concentrate on three distinct types of intermediate steps: document summary, document outline, and document key information.

**Document Summary**   A document summary is a concise representation that captures the core message of a document, summarizing key points, themes, or arguments while omitting peripheral details. It provides a quick overview, enabling readers to grasp the essential content without reading the entire document.

**Document Outline**   A document outline represents the hierarchical structure of a document. It reveals the organization and flow of ideas, the relationships between sections, and the key points within the document.

**Document Key Information**   Document key information includes the most crucial facts, findings, or insights within a document. It typically represents the core knowledge that the document aims to convey or support.

### 4.2 STRUCTURED INTERMEDIATE STEPS CONSTRUCTION

The intermediate steps $\mathbf{z}_i$ involved in the creating long-form documents provide valuable structural guidance for the generation process. However, this structured information is not always explicitly available, nor is the alignment between the structure and the final document. As discussed above, authors typically establish these intermediate steps when writing long-form documents. Therefore, we posit that such implicit structural information is already embedded within the final document itself.

Building on this observation, we propose a method to extract the intermediate steps from established documents $y_i$, such as news articles and Wikipedia entries. Leveraging recent advances in LLMs, particularly their impressive few-shot learning capabilities, we aim to synthetically generate these intermediate steps directly from the documents. This approach captures the implicit structural information without relying on explicitly provided intermediate steps. Furthermore, it can generate multiple intermediate steps for a single document, thus enabling exploration of various organizational

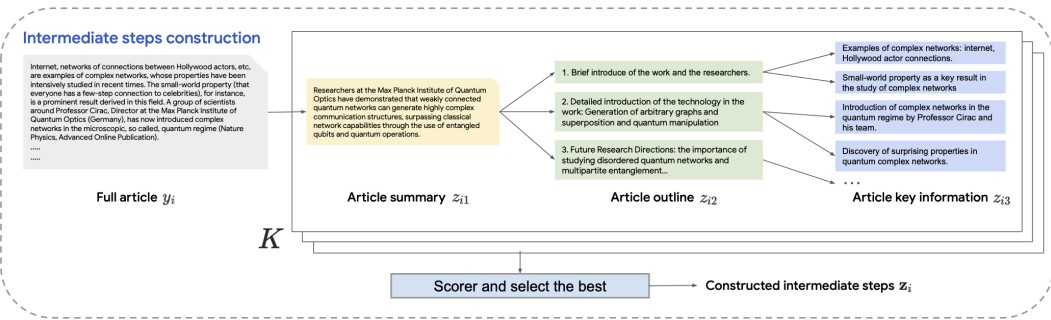

Figure 2: Constructing intermediate steps.

strategies. Lastly, our method provides a consistent and scalable solution for generating intermediate steps, making it well-suited for large-scale document processing. Figure 2 summarizes the process of our proposed process to construct intermediate steps from the article.

To extract the intermediate steps from an article $y_i$, we prompt pre-trained LLMs, i.e., Gemini Ultra, to generate $K$ intermediate step candidates $\{\mathbf{z}_i^k\}_{k=1}^K$, where $\mathbf{z}_i$ can be any kinds of structure information including document summary, document outline, and document key information. Despite LLM's impressive summarization capabilities, they can sometimes produce hallucination information. To mitigate this, we select the best intermediate step from $K$ candidates using a designed scoring function that evaluate coherence and completeness with respect to the document $y_i$. The candidate that demonstrates the highest overall quality is chosen.

The construction of the intermediate steps is detailed in Algorithm 1. Take the document summary as an example, we first normalize the word and paragraph counts relative to the input text. We then apply a sinusoidal scoring function, $\mathbf{g}(\cdot)$ in the Algorithm 1, derived from empirical observations, to assess the adherence to desirable length and structure. Additionally, we incorporate an entailment score using a hallucination model, measuring the bi-directional logical coherence between the summary and the original news article. In our experiment, we use the hallucination model introduced in Shen et al. (2023). This entailment score assessed both the extent to which the summary accurately reflected the news (no hallucination) and the degree to which the summary covered the key information in the news. These individual scores (word/paragraph count, entailment) were combined multiplicatively to produce a final score for each summary, allowing us to select the highest-scoring option as the optimal summary.

---

**Algorithm 1** Constructing intermediate steps data from organic long-form text data

---

**Input:** Organic long-form text $\mathcal{Y}$
**Output:** Intermediate steps
**for** $y_i$ in $\mathcal{Y}$ **do**
$\quad \{\mathbf{z}_i^k\}_{k=1}^K = \text{few\_shot\_llm}(y_i)$
$\quad$ **for** $\mathbf{z}_i^k$ in $\{\mathbf{z}_i^k\}_{k=1}^K$ **do**
$\quad\quad \text{WordRatio}_k = \text{WordCount}(\mathbf{z}_i^k)/\text{WordCount}(y_i)$
$\quad\quad \text{SentenceRatio}_k = \text{SentenceCount}(\mathbf{z}_i^k)/\text{SentenceCount}(y_i)$
$\quad\quad \text{LengthScore}_k = \mathbf{g}(\text{WordRatio}_k, \text{SentenceRatio}_k)$
$\quad\quad \text{EntailmentScore}_k = \text{HallucinationScore}(y_i, \mathbf{z}_i^k) + \text{HallucinationScore}(\mathbf{z}_i^k, y_i)$
$\quad\quad \text{QuanlityScore}_k = \text{LengthScore}_k \times \text{EntailmentScore}_k$
$\quad$ **end for**
$\quad \mathbf{z}_i = \arg\max_{k=1}^K \text{QualityScore}_k$
**end for**

---

### 4.3 FINE-TUNING

To empower the LLM with the ability to utilize structural information during long-form text generation, we propose a fine-tuning approach that goes beyond the conventional input-to-output (e.g.,

$x_i \to y_i$) paradigm. Our method leverages intermediates steps $\mathbf{z}$, which encapsulate the structural essence of the desired output.

Specifically, we introduce two auxiliary fine-tuning tasks in addition to the standard $x_i \to y_i$ task.

**Input to output with intermediate steps ($x_i \to \mathbf{z}_i \oplus y_i$)** : This task trains the LLM to generate both the final long-form text $y_i$ and the corresponding intermediate steps $\mathbf{z}_i$. It encourages the model to internalize the relationship between content and structure, fostering a deeper understanding of how different elements of a document contribute to its overall organization. For this task, the training dataset is,

$$D = \{(x_1, \mathbf{z}_1 \oplus y_1), (x_2, \mathbf{z}_2 \oplus y_2), \ldots, (x_n, \mathbf{z}_n \oplus y_n)\}$$

**Input and intermediate steps to output ($x_i \oplus \mathbf{z}_i \to y_i$)** : In this task, the LLM is provided with both the input $x$ and the synthetically generated intermediate steps $\mathbf{z}_i$ as context. The model is then trained to generate the final long-form text $y_i$, conditioned on the structural guidance provided by $\mathbf{z}$. Here, $\mathbf{z}$ acts as a blueprint for generating the long-form text, helping the model maintain focus and generate more coherent and structured output. In this task, the training dataset is:

$$D = \{(x_1 \oplus \mathbf{z}_1, y_1), (x_2 \oplus \mathbf{z}_2, y_2), \ldots, (x_n \oplus \mathbf{z}_n, \oplus y_n)\}$$

The two tasks complement each other by focusing on different aspects of the planned generation process. The $(x_i \to \mathbf{z}_i \oplus y_i)$ task teaches the model to generate structure alongside content, emphasizing the inherent relationship between the two. Meanwhile, the $(x_i \oplus \mathbf{z}_i \to y_i)$ task trains the model to effectively utilize provided structural information, improving its ability to follow a given plan. By incorporating these auxiliary tasks, our fine-tuning process aims to improve the LLM's capability of generating long-form text that is not only informative but also well-structured.

## 5 EXPERIMENTS

We conduct extensive experiments on multiple datasets with different setups to validate the effectiveness of our proposed methods.

### 5.1 BENCHMARK DATASETS

To validate the effectiveness of our planned generation method, we conducted experiments using the SciNews dataset and Wikipedia dataset.

**SciNews Dataset.** SciNews dataset (Pu et al., 2024) is developed to facilitate the task of compiling academic publications into scientific news reports. This dataset provides a parallel collection of academic publications and their corresponding scientific news reports across 9 disciplines, including Medicine, Biology, Physics, and Chemistry. It comprises 41,872 samples, with each academic publication averaging 7,760.90 tokens and each news report averaging 694 tokens. In this dataset, we use each academic publication as the input $x_i$ and its corresponding news report as $y_i$. We randomly sample 33,497 items as the training split, 1,000 items for validation and 200 items as the evaluation split.

**Wikipedia Dataset.** We use the Wikipedia corpus from the KILT benchmark dataset (Petroni et al., 2020) to construct the training and validation splits. This benchmark dataset is sourced from a snapshot of Wikipedia taken on August 1, 2019, and contains 5.9 million articles. The evaluation split is constructed from the FreshWiki corpus (Shao et al., 2024), which consists of 100 high-quality Wikipedia page with most edits for each month from February 2022 to September 2023. Both dataset includes the full text of Wikipedia pages, along with descriptions and titles. We filtered both corpus by removing articles with less than 1,000 words and those lacking any structured sections.

For each Wikipedia article $y_i$, we formulated the input context $x_i$ as a one sentence prompt:

*Generate a comprehensive Wikipedia page about the specified topic. topic: [entity]*

We randomly sampled 1,900 items from the KILT corpus as the training split, 232 items for the validation split, and used all the 98 filtered items from the FreshWiki corpus as the evaluation split.

## 5.2 EVALUATION METRICS

**Offline metrics.** To assess the effectiveness of the proposed method, we utilized the F1 scores of ROUGE-1 (R1), ROUGE-2 (R2), ROUGE-L (RL), and ROUGE-Lsum (RLsum) metrics for this evaluation.

**Human and auto SxS.** In addition to these ROUGE metrics, and in line with the concept of critical evaluation (Xu et al., 2023b), we also conducted SxS evaluations, both by human expert raters and by an LLM evaluator. For the LLM evaluator, we employed a more capable LLM (Gemini Ultra) as an automatic SxS rater to compare the generated articles from two methods. For both human and auto SxS, raters are asked to rate which one of two generated articles is better. The base and the test sides are randomly flipped to avoid potential bias towards one side. We calculate and report the win rate and W/L ratio of our proposed method compared against the baseline.

## 5.3 EXPERIMENT SETUP

We use Gemini Pro model for both our baselines and all of our fine-tuning experiments. For experiments on the SciNews dataset, we limit the input sequence length to the model to be no more than around 16k tokens, and the output sequence length to be no more than 4k tokens. For experiments on the Wikipedia dataset, we limit the input sequence length to be no more than 1k tokens as the input context in this data set is much shorter. The output sequence length limit is set to 6k tokens.

All the fine-tuning experiments are conducted on TPU. The batch size is set to 16 and 32 respectively for SciNews and Wikipedia datasets. The maximum learning rate for all experiments is set as $10^{-5}$.

## 5.4 RESULTS

**Overall comparison.** Table 1 shows the results of the following models and prompt settings:

1. **Zero-shot (ZS):** Directly using the LLM without fine-tuning. The input context $x_i$ to the LLM is the same as the our proposed methods.

2. **Fine-tuning without intermediate steps (FT w/o I):** Only fine-tune the LLM with the input context $x_i$ as input and the output article $y_i$ as output without using any intermediate steps $\mathbf{z}_i$.

3. **Fine-tuning with intermediate steps (FT w/ I):** Fine-tune the LLM with the a mixture of two training tasks: generating the full article $y_i$ from the input context $x_i$ ($x_i \rightarrow y_i$), and generating all the intermediate steps $\mathbf{z}_i$ and the full article $y_i$ from the input context $x_i$ ($x_i \rightarrow \mathbf{z}_i \oplus y_i$). The instruction prompt from the two tasks are different to ensure the LLM can differentiate these tasks. See Section 4.3 for more details. During inference the LLM is prompted with input context $x_i$ to only output the full article $y_i$ ($x_i \rightarrow y_i$).

4. **Fine-tuning and inference with intermediate steps (FT w/ I, Output w/ I):** Similar to the fine-tuning with intermediate steps method above, except that during inference the LLM is prompted with input context $x_i$ to generate both intermediate steps $\mathbf{z}_i$ and the full article $y_i$ ($x_i \rightarrow \mathbf{z}_i \oplus y_i$). Only the full article is extracted from the generated output for evaluation.

Table 1: Performance of different methods with various training and prompt setups

| Dataset | Methods | $\mathbf{R1}_{F1}$ | $\mathbf{R2}_{F1}$ | $\mathbf{RL}_{F1}$ | $\mathbf{RLsum}_{F1}$ |
|---|---|---|---|---|---|
| SciNews | ZS | 37.16 | 10.03 | 16.30 | 35.03 |
| | FT w/o I | 46.66 | 13.39 | 19.26 | 44.14 |
| | FT w/ I | 48.63 | 14.39 | 19.57 | 45.92 |
| | FT w/ I, Output w/ I | **49.39** | **14.69** | **19.68** | **46.61** |
| Wikipedia | ZS | 28.51 | 10.65 | 13.91 | 27.52 |
| | FT w/o I | 42.61 | 13.69 | 16.30 | 41.28 |
| | FT w/ I | 45.65 | 15.21 | 17.44 | 44.23 |
| | FT w/ I, Output w/ I | **47.07** | **15.72** | **17.72** | **45.67** |

Compared with the zero-shot baseline ZS, fine-tuning LLM, even without the intermediate steps, significantly improved the ROUGE scores. Moreover, when LLM is fine-tuned with the mixture training data which incorporated the task of generating intermediate steps, the performance is further improved on both datasets. Specifically, on both datasets, the R1 score improved by more than +2.7-4.4% when compared to the FT w/o I baseline, the R2 score by more than +1.3-2.0%, and the RLsum score by more than +2.5-4.4%.

These results demonstrate that the inclusion of the generating intermediate steps task during fine-tuning significantly enhances long-form text generation by LLMs. The improvements across various metrics and both datasets highlight the robustness of the proposed methods.

We also observe that prompting the LLM to output the intermediate steps $\mathbf{z}_i$ during inference (FT w/I, Output W/I) achieves better performance when prompting the LLM to only output the article (FT w/I). This is expected as explicitly writing down the planning process during the inference can help the model to generate more structured article, similar to the usage of chain-of-thought prompting for reasoning tasks (Wei et al., 2022).

**Comparison of different training and inference recipes.** We also conduct a comparison to understand whether different combinations of training data mixtures and inference tasks lead to different effectiveness. Each mixture contains one or multiple training tasks: directly generating the article from input context $x_i \rightarrow y_i$; generating both the intermediate steps and the article from the input context $x_i \rightarrow \mathbf{z}_i \oplus y_i$; and generating the article from the input context and the intermediate steps: $x_i \oplus \mathbf{z}_i \rightarrow y_i$. We also test different inference tasks, including only prompting the LLM to output the entire article ($x_i \rightarrow y_i$) or prompting the LLM to output both the intermediate steps and the article ($x_i \rightarrow \mathbf{z}_i \oplus y_i$).

Table 2: Performance of models with different training mixtures during fine-tuning.

| Dataset | Training Mixture | Inference | $\mathbf{R1}_{F1}$ | $\mathbf{R2}_{F1}$ | $\mathbf{RL}_{F1}$ | $\mathbf{RLsum}_{F1}$ |
|---|---|---|---|---|---|---|
| SciNews | $x_i \rightarrow y_i$ | $x_i \rightarrow y_i$ | 46.66 | 13.39 | 19.26 | 44.14 |
| | $x_i \rightarrow \mathbf{z}_i \oplus y_i$ | $x_i \rightarrow y_i$ | 48.94 | 14.28 | 19.48 | 45.54 |
| | $x_i \rightarrow \mathbf{z}_i \oplus y_i$ | $x_i \rightarrow \mathbf{z}_i \oplus y_i$ | 49.02 | 14.50 | 19.61 | 46.24 |
| | $x_i \rightarrow y_i$; $x_i \rightarrow \mathbf{z}_i \oplus y_i$ | $x_i \rightarrow y_i$ | 48.63 | 14.39 | 19.57 | 45.92 |
| | $x_i \rightarrow y_i$; $x_i \rightarrow \mathbf{z}_i \oplus y_i$ | $x_i \rightarrow \mathbf{z}_i \oplus y_i$ | **49.39** | **14.69** | **19.68** | **46.61** |
| | $x_i \rightarrow y_i$; $x_i \rightarrow \mathbf{z}_i \oplus y_i$; $x_i \oplus \mathbf{z}_i \rightarrow y_i$ | $x_i \rightarrow y_i$ | 47.99 | 14.21 | 19.42 | 45.33 |
| Wikipedia | $x_i \rightarrow y_i$ | $x_i \rightarrow y_i$ | 42.61 | 13.69 | 16.30 | 41.28 |
| | $x_i \rightarrow \mathbf{z}_i \oplus y_i$ | $x_i \rightarrow y_i$ | 35.49 | 12.58 | 15.06 | 34.28 |
| | $x_i \rightarrow \mathbf{z}_i \oplus y_i$ | $x_i \rightarrow \mathbf{z}_i \oplus y_i$ | 44.04 | 14.39 | 16.98 | 42.69 |
| | $x_i \rightarrow y_i$; $x_i \rightarrow \mathbf{z}_i \oplus y_i$ | $x_i \rightarrow y_i$ | 45.65 | 15.21 | 17.44 | 44.23 |
| | $x_i \rightarrow y_i$; $x_i \rightarrow \mathbf{z}_i \oplus y_i$ | $x_i \rightarrow \mathbf{z}_i \oplus y_i$ | **47.07** | **15.72** | **17.72** | **45.67** |
| | $x_i \rightarrow y_i$; $x_i \rightarrow \mathbf{z}_i \oplus y_i$; $x_i \oplus \mathbf{z}_i \rightarrow y_i$ | $x_i \rightarrow y_i$ | 45.31 | 14.80 | 17.10 | 43.92 |

Table 2 illustrates the comparison results. The results show that models fine-tuned on the training mixture with intermediate steps consistently outperform models only fine-tuned to generate the article directly. On both datasets, the best performance is achieved when the model is fine-tuned on the mixture of $x_i \rightarrow y_i$ task and the $x_i \rightarrow \mathbf{z}_i \oplus y_i$ task, and prompted to output both the intermediate steps and the article ($x_i \rightarrow \mathbf{z}_i \oplus y_i$) during inference.

We also experiment with the mixture of three tasks on the Wikipedia data set: the vanilla $x_i \rightarrow y_i$ task, the $x_i \rightarrow \mathbf{z}_i \oplus y_i$ task which generates the intermediate steps, and the $x_i \oplus \mathbf{z}_i \rightarrow y_i$ task which generates the article from both the input context and the intermediate steps. However, further adding the third task does not seem to bring extra performance improvement. This might suggest that writing an article given the outlines is a relatively trivial task for the underlying LLM in our experiments. Further fine-tuning the LLM on this task does not bring additional insight for the model to write better article.

Table 3: Single-turn vs. multi-turn inference on SciNews.

| Training Mixture | Inference | $\mathbf{R1}_{F1}$ | $\mathbf{R2}_{F1}$ | $\mathbf{RL}_{F1}$ | $\mathbf{RLsum}_{F1}$ |
|---|---|---|---|---|---|
| $x_i \rightarrow y_i$; $x_i \rightarrow \mathbf{z}_i \oplus y_i$ | $x_i \rightarrow \mathbf{z}_i \oplus y_i$ | **49.39** | **14.69** | **19.68** | **46.61** |
| $x_i \rightarrow y_i$; $x_i \rightarrow \mathbf{z}_i \oplus y_i$ | $x_i \rightarrow \mathbf{z}_i, x_i \oplus \mathbf{z}_i \rightarrow y_i$ | 46.76 | 14.68 | 19.43 | 43.73 |

**Single-turn vs. multi-turn.**  In addition to identifying the best recipe of training mixture and *single-turn* inference, we also compare the most competitive training recipe in Table 2 against multi-turn inference on SciNews. Results in Table 3 show that single-turn inference notably outperforms multi-turn inference.

Table 4: Human SxS results comparing the proposed FT w/ I method to FT w/o I baseline

| Dataset | Overall W/L | Coherence & Organization | Relevance & Focus | Verifiability |
|---|---|---|---|---|
| SciNews | 3.60 | 4.25 | 3.00 | 7.75 |
| Wikipedia | 1.56 | 1.10 | 1.38 | 1.40 |

**Human and auto SxS results.**  To further validate the effectiveness of our methods, we conducted two sets of side-by-side ratings, one by expert human raters and one by LLM autorater, to compare the outputs of our methods against those of the baseline.

For the SciNews dataset, we select the FT w/I method using the training mixture of $x_i \rightarrow y_i$ and $x_i \rightarrow \mathbf{z}_i \oplus y_i$ tasks as it demonstrated the best performance. For the Wikipedia dataset, we select the FT w/I method using the training mixture of three tasks: $x_i \rightarrow y_i$, $x_i \rightarrow \mathbf{z}_i \oplus y_i$ and $x_i \oplus \mathbf{z}_i \rightarrow y_i$, as it shows comparable performance to the method using only two mixtures. In both cases, the baseline is the vanilla FT w/o I method without using intermediate steps.

For human SxS, we asked expert human raters to rate 50 items of each data set. They were asked to compare the two outputs for each input item ("paper body" for SciNews, "topic" for Wikipedia) in three of the criteria defined in Shao et al. (2024) that are most relevant for our task, namely *coherence and organization*, *relevance and focus*, and *verifiability*, and to provide an overall assessment. The results are presented in Table 4. Our methods show strong win against the baseline on SciNews, both in the overall quality and in each of the three criteria. On Wikipedia, we also observed a positive result in all criteria and overall.

Besides Human SxS evaluation, we also employed the automated LLM SxS as additional validation, on 200 samples from SciNews and all 100 samples of FreshWiki. The results show that the FT w/I method achieves W/L ratio of 2.85 and 1.20 on SciNews and Wikipedia datasets, respectively, both larger than 1.

Table 5: Performance of different strategies on SciNews with Gemini 1.5 Flash as the base model.

| Training Mixture | Inference | $\mathbf{R1}_{F1}$ | $\mathbf{R2}_{F1}$ | $\mathbf{RL}_{F1}$ | $\mathbf{RLsum}_{F1}$ |
|---|---|---|---|---|---|
| zero-shot | $x_i \rightarrow y_i$ | 43.93 | 11.61 | 17.87 | 41.22 |
| zero-shot | $x_i \rightarrow \mathbf{z}_i \oplus y_i$ | 40.62 | 10.39 | 17.20 | 38.18 |
| $x_i \rightarrow y_i$ | $x_i \rightarrow y_i$ | 45.38 | 12.11 | 18.00 | 42.62 |
| $x_i \rightarrow y_i; x_i \rightarrow \mathbf{z}_i \oplus y_i$ | $x_i \rightarrow y_i$ | 46.32 | 12.32 | 18.05 | 43.48 |
| $x_i \rightarrow y_i; x_i \rightarrow \mathbf{z}_i \oplus y_i$ | $x_i \rightarrow \mathbf{z}_i \oplus y_i$ | **46.53** | **12.45** | **18.16** | **43.77** |

**Alternative base model: Gemini 1.5 Flash.**  To show that our method are applicable to the latest generation of LLMs, we also conducted smaller scale experiments (due to cost limitations) on Gemini 1.5 Flash as the base model. Table 5 show the results on SciNews, which corroborate the findings on Gemini 1.0 Pro shown in Table 2.

## 6 CONCLUSIONS

In this paper, we explore to fine-tune LLM to write long-form articles in one single turn. We propose to include intermediate planning steps, such as starting with a concise summary, writing down the outlines and collecting some key information into the mixture of the training data. Noting that such intermediate steps are not available in most existing data sets, we propose to construct synthetic intermediate steps from existing full-length articles. We prompt an LLM to extract, shorten and summarize the article into a tree structure, where each level corresponds to an intermediate step. We fine-tune the LLM writer with different mixtures before prompt the LLM writer to write the full article. Our experiments on two data sets from different domains: SciNews and Wikipedia, verify that our proposed method can substantially boost the quality of the generated articles.

Our exploration creates a new paradigm in long-form text generation: instead of applying LLM iteratively to generate intermediate steps until the full article is obtained, we include the intermediate steps into a mixture of training data and directly fine-tune an LLM to generate all of them in one turn. While we only experiment with one specific way of constructing the intermediate steps (as tree-structured planning outlines), there are many other different possible ways to create the auxiliary training mixture. As modern LLMs support longer and longer input and output sequence lengths, we believe it is possible to build even more more sophisticated long-form LLM writers with embedded capabilities other than planning. For example, one can also fine-tune the LLM to perform fact-checking reasoning on-the-fly to improve the factuality of the generated article.

## 7 LIMITATIONS

In this work we focus on writing long-form articles in a totally closed-book setup, where the only external information is provided in the input context. However, a more realistic setting would be retrieval-augmented generation (RAG), which equips the LLMs with a retriever to external knowledge. Since we do not adopt a RAG setting, we also do not compare to other existing methods that adopt RAG (e.g., STORM (Shao et al., 2024)) While we do not adopt a RAG setting, we believe our work can be easily adapted to incorporate RAG to achieve better performance.

## 8 ETHICS STATEMENT

Our approach to incorporating planning steps into LLM training aims to significantly improve the quality of long-form text generation. This advancement has the potential to benefit various sectors by providing more coherent, accurate, and contextually rich content. We believe that the benefits of our work, including the enhancement of educational resources and the facilitation of better information dissemination, outweigh the minimal risks involved.

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

## A PROMPTS

In this section we provide the prompts we used for different tasks.

### A.1 SCINEWS

**Only generating the full article** ($x_i \rightarrow y_i$). Below we provide the prompt for

> Given the body of the academic paper, generate a whole news article.
> Academic paper body: {input_paper}

**Generating the intermediate steps and the full article ($x_i \rightarrow \mathbf{z}_i \oplus y_i$).** Below we provide the prompt for

> Given the academic paper's full text, first generate a news article's summary, the high-level outline and detailed key information snippets, then leverage those information to generate a complete news article with title and body.
> Academic paper body: {input_paper}

## A.2 WIKIPEDIA

**Only generating the full article ($x_i \rightarrow y_i$).** Below we provide the prompt for

> Generate a comprehensive Wikipedia page about the specified topic.
> Topic: {input_topic}

**Generating the intermediate steps and the full article ($x_i \rightarrow \mathbf{z}_i \oplus y_i$).** Below we provide the prompt for

> Given a specific topic, you are asked to write a comprehensive Wikipedia page about this topic. Let's write step by step. First generate a summary, a high-level outline and a list of detailed key information snippets. Then, follow the summary, high-level outline and detailed key information snippets, generate a Wikipedia page about this topic.
> Topic: {input_topic}

