# OpenReview forum: "Integrating Planning into Single-Turn Long-Form Text Generation"
_ICLR.cc/2025/Conference — Submitted to ICLR 2025_

### Official Review · Reviewer_SbAJ · 2024-11-01

**Soundness:** 3
**Presentation:** 3
**Contribution:** 2
**Rating:** 5
**Confidence:** 4

**Summary:**

This paper proposes a training approach that enhances long-form writing capabilities in LLMs by incorporating pre-writing stages through auxiliary training tasks. In addition to the conventional end-to-end training for direct output generation, the approach includes two intermediate tasks: generating an outline from the input and creating a complete article based on both the input and outline. The model is trained using a mixture of these tasks. During inference, the model can either generate a full article directly or follow the intermediate steps before producing the final output. The method is evaluated on the SciNews and Wikipedia datasets, with both automated metrics (ROUGE and LLM evaluations) and human assessments demonstrating that LLMs fine-tuned with these auxiliary tasks deliver improved outputs.

[After rebuttal] After reading the author responses and other reviewers' comments, I tend to keep my score unchanged.

**Strengths:**

- This paper studies an important task of long-form text generation. Incorporating auxiliary tasks to improve long-form writing performance is interesting;
- Both automatic and human evaluations on two datasets prove the effectiveness of the method;
- The analysis, particularly the findings in Table 2, is interesting and may offer insights to future research.

**Weaknesses:**

- The proposed approach is similar to [1], which also employs multitask training for long-form generation. Clarifying the distinctions between the two methods would enhance the paper;
- Only one LLM (Gemini) is used in the experiments. Including recent open-source LLMs, such as LLAMA3, would strengthen the results' validity, especially as fine-tuning smaller LLMs is often more feasible in practical applications;
- Evaluation is another concern. ROUGE primarily measures word overlap; including more advanced metrics, such as BERTScore, would provide a deeper assessment. Additionally, LLM-based evaluations currently focus on overall quality, lacking fine-grained aspects. Faithfulness and factuality are also absent;
- Sample outputs and error analysis are not provided, which limits understanding of common issues and qualitative insights.



[1] MOCHA: A Multi-Task Training Approach for Coherent Text Generation from Cognitive Perspective, EMNLP'22

**Questions:**

- Is there an analysis of output length across different generation methods?
- Has the quality of the generated outlines been evaluated?
- For human evaluations, what is the inter-annotator agreement score?

---

> ### Author Response · Authors · 2024-12-01
>
> **Weakness 1**: “The proposed approach is similar to [1], which also employs multitask training for long-form generation. Clarifying the distinctions between the two methods would enhance the paper”
>
> **Response 1**:
> Thank you for your insightful suggestions. We appreciate the opportunity to clarify the distinctions between our approach and that of [1].
> 1. **Focused Task Selection in Multi-task Co-training**: While [1] employs multitask training across a variety of related tasks, our method specifically concentrates on tasks designed to train the model to learn from the thinking process. We achieve this by utilizing Chain-of-Thought (CoT) data, including data in the format of summaries, outlines, and key information, to guide the model's reasoning during training. This targeted approach is particularly effective in enhancing the model's ability to generate coherent and logical long-form content. Unlike [1], which uses a range of related tasks, we have a systematic method for designing targeted tasks that demonstrably improve long-form generation through structured reasoning.
> 2. **Novel Method for Generating High-Quality Synthetic CoT Data**: We have developed an innovative method for generating high-quality synthetic CoT data for training, which is not addressed in [1]. This approach allows us to augment our training dataset with rich, synthetic reasoning processes, further enhancing the model's performance in long-form generation tasks. By synthetically generating data tailored to our designed tasks, we ensure that the model is exposed to diverse and informative reasoning patterns during training.
>
> We will ensure that these distinctions are clearly articulated in the revised manuscript.
>
>
>
> **Weakness 2**: Only one LLM (Gemini) is used in the experiments. Including recent open-source LLMs, such as LLAMA3, would strengthen the results' validity, especially as fine-tuning smaller LLMs is often more feasible in practical applications;
>
> **Response 2**: While our experiments focused on Gemini family models due to restrictions on access and resources, we did not leverage any Gemini-specific model capabilities or knowledge. Hence our proposed methods can be seamlessly applied on any other LLMs.
>
>
> **Weakness 3**: Evaluation is another concern. ROUGE primarily measures word overlap; including more advanced metrics, such as BERTScore, would provide a deeper assessment. Additionally, LLM-based evaluations currently focus on overall quality, lacking fine-grained aspects. Faithfulness and factuality are also absent;
>
> **Response 3**: Thank you for highlighting this important concern.
>
> We agree that token-overlapping metrics like ROUGE and LLM-based evaluations have limitations in fully capturing the quality of generated content. To address this, we supplemented our evaluations with comprehensive human assessments, as shown in Table 4. In the human evaluation, we examined multiple dimensions, including **Verifiability** (which helps evaluate **faithfulness and factuality**), Coherence & Organization, Relevance & Focus, and overall performance. Our method demonstrated consistent gains across all these aspects, providing a more nuanced and thorough evaluation beyond what automatic metrics can offer.

---

> ### Author Response · Authors · 2024-12-01
>
> **Question 1**: Is there an analysis of output length across different generation methods?
>
> **Answer 1**: Please find the average output lengths in terms of token counts below:
> - SciNews, baseline: 817
> - SciNews, our method: 830
> - Wikipedia, baseline: 3371
> - Wikipedia, our method: 3399
>
> While our method does produce slightly longer outputs than the baseline, the difference (0.8%-1.6%) is not significant enough to be attributed for the noticeable gains measured by both ROUGE scores and SxS ratings.
>
> **Question 2**: Has the quality of the generated outlines been evaluated?
>
> **Answer 2**:
> The quality of the generated outlines has been evaluated using various classifiers and metrics, as detailed in Section 4.2 and Algorithm 1. We didn’t conduct any human evaluations for this aspect as outline generation is not the primary focus of this paper.
>
> The **primary aim of this work** is to improve performance in end-to-end tasks, specifically long-form text generation. Our findings indicate that high-quality CoT data can significantly enhance reasoning capabilities in writing.
>
> **To meet this need**, we developed novel methods to synthetically generate high-quality CoT-type data, including summaries, **outlines**, and key information.
>
> These datasets, including outlines, function as intermediate CoT data, designed to improve the performance of the end-to-end task. Training with these datasets has **demonstrated effectiveness** in enhancing the model's reasoning abilities and boosting the final performance of the news generation task, as evidenced in Tables 1–5.
>
>
> **Question 3**: For human evaluations, what is the inter-annotator agreement score?
>
> **Answer 3**: For human eval results, here are the 95% confidence intervals for the W/L ratios:
>
> | | SciNews | Wikipedia |
> | -------- | ------- |------- |
> | Overall W/L | (1.94, 9.34) | (0.80, 3.35) |
> | Coherence & Organization | (2.16, 13.45) | (0.56, 2.19) |
> | Relevance & Focus | (1.53, 8.04) | (0.67, 3.03) |
> | Verifiability | (3.18, 149.16) | (0.66, 3.22) |
>
> On SciNews our method shows a clear advantage over the baseline.  On Wikipedia, while the results are inconclusive, they are leaning positive / in favor of our method.  We’ll include these results in the revision of the paper.

---

### Official Review · Reviewer_EFM6 · 2024-11-01

**Soundness:** 2
**Presentation:** 3
**Contribution:** 2
**Rating:** 3
**Confidence:** 4

**Summary:**

In this paper, the authors propose integrating planning into single-turn long-form text generation using Large Language Models (LLMs). The approach involves an auxiliary task, generating intermediate steps, to enhance coherence and structure in the finally generated text. The authors construct training data by leveraging LLMs to generate synthetic intermediate writing data based on the full articles. Experiments demonstrate improved quality and coherence in long-form documents generated by LLMs fine-tuned with this method.

**Strengths:**

Pros:
* The paper is well-written and easy to follow.
* The construction of the training data is automatic, leveraging off-the-shelf LLMs to generate synthetic intermediate writing data based on the full articles, which saves the expensive cost of manual annotation.
* With the auxiliary task at training time, the model can learn to generate more coherent and structured long-form text in a single pass.

**Weaknesses:**

Cons:
* The paper lacks a comparison with previous work like ProGen (Tan et al., 2021), Ex$^3$ (Huang et al., 2024), etc. The invloved baseline training and prompt setups are too simple to demonstrate the superiority of the proposed approach.
* Besides the cohenrence and structure, the approach in this paper appears to be more efficient and faster than the previous work. The paper lacks metrics to measure the efficiency and speed of the proposed approach compared to previous work.
* The offline metrics used in the paper are not sufficient to evaluate the *cohenrence* and *structure* of the generated text.

References:
[1] Tan, Bowen, et al. "Progressive Generation of Long Text with Pretrained Language Models." Proceedings of the 2021 Conference of the North American Chapter of the Association for Computational Linguistics: Human Language Technologies. 2021.
[2] Lei, Huang, et al. "Ex3: Automatic Novel Writing by Extracting, Excelsior and Expanding." Proceedings of the 62nd Annual Meeting of the Association for Computational Linguistics (Volume 1: Long Papers). 2024.

**Questions:**

1. In line 38-40, the author mentioned that the proposed approach enables us to fully leverage the token-level attention mechanism in the decoding process. How does the token-level attention mechanism is fully used in the decoding process? How does it ensure the coherence and consistency of the generated document? These are not clear to me.
2. Compared to the previous work involving multi-turn generation passes, does the proposed approach have any advantages in terms of the length of the generated text? Can the proposed approach generate longer and better text than the previous work?

---

> ### Author Response · Authors · 2024-11-19
> **Offline evaluation**
>
> We’d like to thank the reviewer for their review.
>
> Regarding Cons #3: "The offline metrics used in the paper are not sufficient to evaluate the cohenrence and structure of the generated text."
>
> We’d like to note that besides ROUGE scores, we also reported in Table 4 the human SxS results by expert human raters, comparing our method against the baseline.  In the table, we show that our method outperformed the baseline on both SciNews and Wikipedia datasets, in three criteria namely *coherence and organization*, relevance and focus, and verifiability.  These aspects were defined in (Shao et al. 2024) which are most relevant for our task.
>
> As additional validation, we also conducted automated LLM SxS eval, which also showed that our method produced better results than the baseline.
>
> References:
> Shao et al. 2024: Assisting in Writing Wikipedia-like Articles From Scratch with Large Language Models. NAACL 2024.

---

> ### Author Response · Authors · 2024-11-19
> **Baselines**
>
> Cons #1: "The paper lacks a comparison with previous work like ProGen (Tan et al., 2021), Ex^3 (Huang et al., 2024), etc. The invloved baseline training and prompt setups are too simple to demonstrate the superiority of the proposed approach."
>
> We’d like to thank the reviewer for pointing out these related works.
>
> While ProGen also tackles the long text generation with a progressive generation mechanism, it adopts a **multi-stage** instead of a single-turn methodology. This involves multiple distinct decoding processes for each stage, where each stage requires its own finetuned BART model and its own decoding vocabulary. While having separate specialist models for different stages can potentially optimize the generation quality, this comes at the cost of increased training complexity and deployment overhead compared to our proposed single-turn approach. Given these fundamental differences in design and resource requirements, a direct comparison with ProGen would be unfair.
>
> For Ex^3, it indeed employed similar ideas to solve a similar problem to ours. We’d respectfully note that it first appeared on Arxiv on Aug 16, 2024, hence shall be considered as concurrent work as ours.

---

### Official Review · Reviewer_vz33 · 2024-11-03

**Soundness:** 3
**Presentation:** 4
**Contribution:** 3
**Rating:** 8
**Confidence:** 5

**Summary:**

This paper proposes to fine-tune LLMs to generate long-form content directly in a single pass through a pre-writing auxiliary task that teaches the LLM to plan, reason and structure. They leverage Gemini Ultra to generate the training data for this auxiliary task, namely generating synthetic summaries, outlines and key informations from full articles. Three training paradigms are used to fine-tune the model. To mitigate the possible hallucinations when generating synthetic data, they have a sinusoidal scoring function to assess the adherence to length and structure, and a hallucination model for coherence between the synthetic data and original article. Improvements are shown in Rouge-LSUM, and 3.6 win/loss ratio in organization, relevance and verifiability.

**Strengths:**

- Very well-written paper! The proposed method is explained in a clear way, with the difference of other objectives (generating in multiple turns) distinguished.
- The proposed method’s intermediate steps are all in scope, and all intuitively should help with full document generation. They also try to mitigate hallucinations by using a score function of coherence and completeness to choose the highest overall quality. As confirmed by both automatic metrics as well as human/LLM as judge metric, the method constantly performs better than the baseline.
- The ablation is relatively thorough. The authors compare against different evaluation/training paradigms, different mixture of training data, and single-turn / multi-turn inference.

**Weaknesses:**

- For evaluation metrics, although human and auto SxS are used, only 50 articles are rated by human. Furthermore, the LLM that is used to generate the synthetic data is used as the rater. As suggests by some previous work (Panickssery et al. 2024), for example, LLM raters might be able to recognize and favor their own generations. Therefore, there are chances that the model that is trained on additional Gemini Ultra generated data might be favored more than zero-shot.
- Length impact — it seems that for SciNews, the news report is 694 tokens, but the LLM is allowed to generate no more than 4k tokens, and for Wikipedia dataset, FreshWiki’s length is less than 3k words, and the LLM is allowed to generate no more than 6k tokens. Although there is no comparison against the groundtruth documents, is it possible that some method generates more/less tokens and thus is favored by either Rouge scores or SxS raters?
- The paper only tested the Gemini family models.

**Questions:**

- Although it might not impact the baseline comparison because there are no comparison against groundtruth data, for Wikipedia dataset, is there any possible data contamination?
- What does multi-turn inference mean in line 432? Is it prompt LLM to generate each intermediate first then generate the full document?
- Why is the coherence & organization for Wikipedia models win ratio bad? Is it because more hallucinations appears? Does this mean that the hallucination mitigation is less effective on Wikipedia data?

---

> ### Author Response · Authors · 2024-11-27
>
> We sincerely appreciate your positive feedback on our work.  We hope our responses below could satisfactorily address your questions.
>
> **Weakness 1**: *“For evaluation metrics, although human and auto SxS are used, only 50 articles are rated by human. Furthermore, the LLM that is used to generate the synthetic data is used as the rater. As suggests by some previous work (Panickssery et al. 2024), for example, LLM raters might be able to recognize and favor their own generations. Therefore, there are chances that the model that is trained on additional Gemini Ultra generated data might be favored more than zero-shot.”*
>
> For human eval results, here are the 95% confidence intervals for the W/L ratios:
>
> | | SciNews | Wikipedia |
> | -------- | ------- |------- |
> | Overall W/L | (1.94, 9.34) | (0.80, 3.35) |
> | Coherence & Organization | (2.16, 13.45) | (0.56, 2.19) |
> | Relevance & Focus | (1.53, 8.04) | (0.67, 3.03) |
> | Verifiability | (3.18, 149.16) | (0.66, 3.22) |
>
> On SciNews our method shows a clear advantage over the baseline.  On Wikipedia, while the results are inconclusive, they are leaning positive / in favor of our method.  We’ll include these results in the revision of the paper.
>
> Thank you for bringing up the related work by Panickssery et al.  We acknowledge the possibility that Gemini Ultra as the autoSxS rater may favor the finetuned model.  The autoSxS results are offered more as additional validation rather than primary evidence.
>
>
>
> **Weakness 2**: *“Length impact — it seems that for SciNews, the news report is 694 tokens, but the LLM is allowed to generate no more than 4k tokens, and for Wikipedia dataset, FreshWiki’s length is less than 3k words, and the LLM is allowed to generate no more than 6k tokens. Although there is no comparison against the groundtruth documents, is it possible that some method generates more/less tokens and thus is favored by either Rouge scores or SxS raters?”*
>
> Here are the average output lengths in terms of token counts:
> - SciNews, baseline: 817
> - SciNews, our method: 830
> - Wikipedia, baseline: 3371
> - Wikipedia, our method: 3399
>
> While our method does produce slightly longer outputs than the baseline, the difference (0.8%-1.6%) is not significant enough to be attributed for the noticeable gains measured by both ROUGE scores and SxS ratings.
>
>
>
> **Weakness 3**: *“The paper only tested the Gemini family models.”*
>
> While our experiments focused on Gemini family models due to restrictions on access and resources, we did not leverage any Gemini-specific model capabilities or knowledge. Hence our proposed methods can be seamlessly applied on any other LLMs.

---

> ### Author Response · Authors · 2024-11-27
>
> **Question 1**: *“Although it might not impact the baseline comparison because there are no comparison against groundtruth data, for Wikipedia dataset, is there any possible data contamination?”*
>
> For Wikipedia, we train on KILT and evaluate on FreshWiki.  The topics in the two corpora are completely disjoint.  The models may possess knowledge about topics in FreshWiki, but the training process doesn’t alter such knowledge for the model, just trains it to compose long articles differently.
>
>
>
> **Question 2**: *“What does multi-turn inference mean in line 432? Is it prompt LLM to generate each intermediate first then generate the full document?”*
>
> Yes, exactly.  Here are the prompts used for multi-turn generation.
>
> ----------------------------------------------
>
> Given the full text of an academic paper, generate a one sentence summary for a news article that covers the key findings, methodologies, and implications of the academic paper.
>
> academic paper body:  {paper_body}
>
> ----------------------------------------------
>
> Given the full text of an academic paper, generate high-level outlines for a news article that covers the key findings, methodologies, and implications of the academic paper.
>
> academic paper body: {paper_body}
>
> ----------------------------------------------
>
> Given the full text of an academic paper, generate detailed key information snippets for a news article that covers the key findings, methodologies, and implications of the academic paper.
>
> academic paper body: {paper_body}
>
> ----------------------------------------------
>
> Given the academic paper's full text, the news article's summary, high-level outline, and detailed key information snippets, generate a complete news article.
>
> academic paper body: {paper_body}
>
> news summary: {summary}
>
> news high-level outline: {outline}
>
> news key information: {key_info}
>
> ----------------------------------------------
>
>
>
>
>
> **Question 3**: *“Why is the coherence & organization for Wikipedia models win ratio bad? Is it because more hallucinations appears? Does this mean that the hallucination mitigation is less effective on Wikipedia data?”*
>
> The W/L ratio of 1.10 for coherence & organization for Wikipedia is still positive, just not as high as that on SciNews (4.25).  Our conjecture is that the base model is more familiar with Wikipedia style writing than paper-->news, hence more room for improvement for the latter.
>
> As for hallucinations, it’s possible that the model hallucinates more on Wikipedia since the input is only the topic, and the models would rely on their internal knowledge in composing the article.  However, hallucinations would negatively affect verifiability more than coherence & organization.  Retrieval augmentation would mitigate hallucination, for both the finetuned model and the base model, but that’s beyond the scope of this work.

---

> ### Comment · Reviewer_vz33 · 2024-12-02
> **Response to the authors**
>
> Hi,
> Thank you for the response, and my questions are answered. Since my score is already high, I will keep the same score.
>
> Thanks!

---

### Official Review · Reviewer_cUDY · 2024-11-03

**Soundness:** 2
**Presentation:** 2
**Contribution:** 2
**Rating:** 3
**Confidence:** 4

**Summary:**

This paper presents an approach to integrate planning into single-turn long-form text generation using Large Language Models (LLMs). The authors propose generating intermediate steps through an auxiliary task that teaches LLMs to plan, reason, and structure content before final text generation. To address the scarcity of training data for intermediate steps, the authors leverage LLMs to create synthetic intermediate writing data from existing full articles. Experiments on the SciNews and Wikipedia datasets demonstrate that fine-tuning LLMs with this auxiliary task results in higher quality documents, with improvements in ROUGE scores and human evaluation metrics.

**Strengths:**

The authors have validated the effectiveness of their method through both automated metrics and human evaluation, showcasing the method's practical utility.

**Weaknesses:**

1. There is significant room for improvement in the writing style and structure of the paper. For example, Sections 4.1, 4.2, and 5.1 are overly verbose and could benefit from a more concise presentation, with implementation details potentially moved to the appendix. The organization of the paper is somewhat disjointed, failing to highlight key points effectively. For instance, the second task introduced in section 4.3 is mentioned as ineffective in the experiments (Table 2); thus, its inclusion in the methodology section seems unnecessary.
2. The methodology lacks novelty. The experiments are limited in scope and content, especially considering the paper could benefit from a more streamlined language.

**Questions:**

1. I am curious about the impact of this training method on the model's performance in generating short texts.
2. If we directly guide the model to first generate intermediate steps and then the full article, how does its effectiveness compare to the FT w/ I, Output w/ I method? Is the reasoning time roughly the same for both? For broader application, I believe the former might be better; however, considering efficiency, FT w/ I could be more suitable.
3. What is the difference between the loss calculated for the second task mentioned in section 4.3 and the first task in terms of the $y_i$ part?

---

> ### Author Response · Authors · 2024-11-27
>
> We’d like to thank the reviewer for their review. Please find our responses below to the issues raised in the review.
>
>
> **Weakness 1**: *“There is significant room for improvement in the writing style and structure of the paper. For example, Sections 4.1, 4.2, and 5.1 are overly verbose and could benefit from a more concise presentation, with implementation details potentially moved to the appendix.”*
>
> We sincerely appreciate the suggestions, and will use them to improve the organization and writing of the paper.
>
> *“The organization of the paper is somewhat disjointed, failing to highlight key points effectively. For instance, the second task introduced in section 4.3 is mentioned as ineffective in the experiments (Table 2); thus, its inclusion in the methodology section seems unnecessary.”*
>
> Unfortunately, including the second task (x+z->y) in the training mixture didn’t improve the model performance.  Nonetheless, we feel this task is a natural design and excluding it would hinder the comprehensiveness of the study.  However we do reckon that it could be organized differently in order to highlight the key contributions.  Thank you for your suggestion.
>
>
> **Weakness 2**: *“The methodology lacks novelty. The experiments are limited in scope and content, especially considering the paper could benefit from a more streamlined language.”*
>
> We’d like to respectfully emphasize that the novelty in our method is twofold:
> 1. High-quality synthetic CoT data generation for training.
> 2. Long-form text generation in a single pass that embodies both performance and efficiency.
>
> They are supported by a variety of experimental results from ROUGE metrics to LLM autoeval and expert human ratings.
>
>
>
> **Question 1**: *“I am curious about the impact of this training method on the model's performance in generating short texts.”*
>
> The motivation of our proposed method is to introduce human-like planning to improve LLM’s performance in application-specific long text generation.  So in that sense short text generation was not an objective of ours.
>
>
>
> **Question 2**: *“If we directly guide the model to first generate intermediate steps and then the full article, how does its effectiveness compare to the FT w/ I, Output w/ I method? Is the reasoning time roughly the same for both? For broader application, I believe the former might be better; however, considering efficiency, FT w/ I could be more suitable.”*
>
> Thanks for the great question. About effectiveness, please see Table 5, row 2 vs. row 5.  Regarding efficiency, let’s use Google Cloud’s pricing [1] as a proxy as the pricing is proportional to input and output token lengths. The average price per item on the SciNews data set is \\$0.0205 for our method, and \\$0.0626 for multi-step inference.  This shows a competitive advantage for our method in terms of inference efficiency.
>
>
>
> **Question 3**: *“What is the difference between the loss calculated for the second task mentioned in section 4.3 and the first task in terms of the y_i part?”*
>
> The loss function is the same for the two tasks, which is negative log likelihood.
>
> [1] https://cloud.google.com/vertex-ai/generative-ai/pricing

---

### Author Response · Authors · 2024-12-02

Dear Reviewers,

We sincerely appreciate your valuable comments and insights. As the deadline for the Author-Reviewer discussion approaches, we look forward to your feedback. Please let us know if there are any additional clarifications or experiments we can provide. We would be happy to address any remaining concerns or discuss further if needed. Thank you once again for your time and consideration.

Best,
Authors

---

### Meta-Review · Area_Chair_a2KH · 2024-12-20

**Metareview:**

The paper addresses long-form text generation by planning intermediate steps. Reviewers gave inconsistent scores. Compliments include being well-written and thorough analysis. Concerns, however, are also raised, regarding the lack of novelty and lack of comparison with previous work.

The authors mention that their work differs from previous work in multiple ways, such as CoT reasoning and single-turn vs. multi-turn. However, I do not feel such difference is substantial enough to warrant a significant contribution or the waiver of experimental comparison.

**Additional Comments On Reviewer Discussion:**

The paper addresses long-form text generation by planning intermediate steps. Reviewers gave inconsistent scores. Compliments include being well-written and thorough analysis. Concerns, however, are also raised, regarding the lack of novelty and lack of comparison with previous work.

The authors mention that their work differs from previous work in multiple ways, such as CoT reasoning and single-turn vs. multi-turn. However, I do not feel such difference is substantial enough to warrant a significant contribution or the waiver of experimental comparison.

---

### Decision · Program_Chairs · 2025-01-22

Reject